# An Assessment of Austrian School Students’ Mental Health and Their Wish for Support: A Mixed Methods Approach

**DOI:** 10.3390/ijerph20064749

**Published:** 2023-03-08

**Authors:** Katja Haider, Elke Humer, Magdalena Weber, Christoph Pieh, Tiam Ghorab, Rachel Dale, Carina Dinhof, Afsaneh Gächter, Thomas Probst, Andrea Jesser

**Affiliations:** 1Department of Psychosomatic Medicine and Psychotherapy, University for Continuing Education Krems, 3500 Krems, Austria; 2Department of Organizational Psychology, Private University of Schloss Seeburg, 5201 Seekirchen am Wallersee, Austria

**Keywords:** mental health, mental illness, wish for support, mixed methods research, adolescents, young adults, school students, Austria

## Abstract

The mental health of school students has been severely impacted by the aftermath of the COVID-19 pandemic. The present study used a mixed methods approach to assess students’ mental health and examine their wishes for support to improve their psychological well-being. We further investigated gender and age group differences in the amount of clinically relevant mental health problems and the roles that mental health and gender had on desired support. Between April and May 2022, a total of 616 Austrian students aged between 14 and 20 participated in a cross-sectional online survey (77.4% female; 19.8% male; 2.8% non-binary) assessing wishes for support regarding mental well-being and mental health indicators (depression: PHQ-9; anxiety: GAD-7; insomnia: ISI; stress: PSS-10; eating disorders: SCOFF; alcohol abuse: CAGE). A wish for support was expressed by 46.6% of the students. Qualitative content analysis revealed that the two most important categories of desired support types were “professional help” and “someone to talk to”. The group of students with a wish for support in general significantly more often showed clinically relevant depression, anxiety, insomnia, eating disorders, or high stress symptoms. Students that wished for professional help significantly more often exceeded the cut-off for clinically relevant depression, anxiety, and high stress. Those who wished for someone to talk to significantly more often exceeded the cut-off for clinically relevant eating disorders. The results indicate a great need for support for young people’s mental health problems and that this need is even more urgent for students.

## 1. Introduction

Since the outbreak of the COVID-19 pandemic in 2019, there have been repeated major changes in the daily lives of school students. Not only did they have to deal with several school closures and switches to distance learning, the measures to contain the spread of the virus also entailed an increase in stress and a decrease in critical protective factors for young people’s mental well-being (e.g., daily structure, contact to peers) [1,2]. The pandemic had worrisome psychological consequences for people in general [3,4,5]. The effects on their mental health appeared as increased insomnia, depression, anxiety, and stress symptoms [6,7,8,9]. In Austria, the mental health of the general population also decreased perceptually during the pandemic [10,11,12,13,14]. Meta-analyses have shown that adolescents especially suffered from significant declines in their mental health [15,16]. In Austria, mental health decreases were particularly apparent among young people as well [11,13]. Qualitative studies of Austrian school students and apprentices reported that young people suffered most from school-related concerns as well as from the pandemic-related restrictions and regulations, and limitations on their social contacts, while precisely those social contacts were perceived as the most important resources during the pandemic [17,18].

In addition to the measures that have affected all Austrians during the lockdowns since the COVID-19 pandemic began in March 2020 (e.g., curfews, contact restrictions, mandatory masking and testing), students have also faced several complete and partial school closures (March/April 2020, November/December 2020, and April 2021) and distance learning [19,20]. During the data collection period in spring 2022, the wearing of masks in critical infrastructure (e.g., supermarkets, health care facilities, public transportation) was still mandatory. The mask requirement in schools was lifted the day before the survey began. In May 2022, the new Omicron variants (BA.4 and BA.5) hit Austria. However, restaurants, hotels, and retail resumed normal business after the delta wave in November 2021, and the lockdown for unvaccinated persons also ended in January 2022. Therefore, COVID-19-related restrictions were already relatively minor for everyday life in spring 2022 [21].

However, even more than two years after the outbreak of the COVID-19 virus, and after an above-described easing of the containment measures, impairments in Austrian adolescents’ mental health were still apparent in spring 2022. In April and May, clinically relevant symptoms of depression (girls: 73%, boys: 44%), anxiety (girls: 57%, boys: 35%), sleeping problems (girls: 34%, boys: 21%), moderate stress (girls: 95%, boys: 81%), and frequent suicidal ideation (girls: 24%, boys: 12%) were prevalent in adolescents between the ages of 14 and 20. Compared to February 2021, an increase in girls’ clinically relevant depression, anxiety, insomnia, and suicidal ideation symptomatic was observed [22]. It has to be kept in mind that these samples were convenience samples, not representative samples, thus the amount of clinically relevant symptoms is possibly overestimated.

Furthermore, qualitative research revealed that in 2021 only 3.3% of Austrian students [17] and 2.6% of Austrian apprentices [18] mentioned professional help (e.g., seeing a psychologist or psychotherapist) as a resource they use, despite the concerning evidence on their mental health [23,24]. Sheffield et al. [25] showed that, among other barriers, adolescents mostly found seeking professional help too expensive, did not know where to find it, or perceived it as too far away. In Austria, unaffordability and confusion about the workings of the funding system were also found to be critical external barriers to seeing a psychotherapist [26].

Despite the continuing high levels of psychological distress and the many barriers to seeking professional help, to date little is known, not only in Austria but in general, about what help young people themselves would like to receive to deal with their burdens. The present study, as part of a research project on the effects COVID-19 had on young people’s mental health [22], therefore focussed on the level of support wished for, the type of desired support as well as on the roles that mental health and gender had on desired support in Austrian school students during spring 2022. Differences in students’ wish for support are expected based on their suffering due to a mental illness or their gender, since gender differences in help-seeking behavior are well-known [27]. Hence, the study explored the types of support that students wished for to improve their psychological well-being. The study also focussed on the gender and age group differences in the amount of clinically relevant mental health problems and on differences in gender, age, and mental health between students with a general wish for support, a wish for professional help as well as a wish for someone to talk to.

## 2. Materials and Methods

### 2.1. Study Design

The online survey had a cross-sectional design which aimed to determine the mental health status and the wish for support of Austrian school students between the ages of 14 and 20. The study was conducted as an exploratory sequential mixed methods design. Results of the qualitative analysis informed the quantitative analysis by providing the support wish categories [28]. Data collection was operated via REDCap (Research Electronic Data Capture) hosted on servers of the University for Continuing Education Krems [29,30].

The data collection period was from the 26 April to the 24 May 2022. During this period the COVID-measures allowed students to participate in regular face-to-face classes at school and there were no restrictions to socialize in place. The study acquired approval from the data protection officer as well as from the Ethics Committee of the University for Continuing Education Krems, Austria (protocol code EK GZ 41/2018–2021) and the procedure followed the Declaration of Helsinki.

### 2.2. Participants

Participant recruitment was carried out online. The invitation including the online survey link to participate in the study was forwarded to the school representatives. To be eligible for the study, participants had to be between 14 and 20 years old, attend school in Austria, and give electronic informed consent. Their participation was voluntary without incentives.

### 2.3. Variables

#### 2.3.1. Wish for Support

Wish for support was assessed with a dichotomous answer (yes/no) to the question “Would you like to receive support to improve your mental well-being?”.

#### 2.3.2. Type of Support (Open Question)

If they answered the above-mentioned wish-for-support-question with “yes”, a more detailed question about the type of support they wanted followed. This open-ended question was not only analyzed qualitatively using qualitative content analysis but the categories that emerged from the qualitative analysis were additionally used in subsequent quantitative and mixed-method analyses. Qualitative categories that were suitable in size for statistical analyses were used to examine differences in the students’ desired types of support regarding their mental health measures and gender.

German-validated versions of the following six standardized validated mental health measures were used:

#### 2.3.3. Depressive Symptoms (PHQ-9)

The Patient Health Questionnaire-9 (PHQ-9) assesses the participants’ depressive symptoms [31]. The nine items, of which the PHQ-9 consists, represent criteria for depression. As the items are scaled from 0 (not at all) to 3 (nearly every day), the PHQ-9 outcomes can result in sum scores from 0 to 27. A cut-off point ≥10 was used for participants aged 18 or older to define clinically relevant depressive symptoms [31], whereas a cut-off ≥11 was used for adolescents between 14 and 17 [32]. The PHQ-9 is validated in German [33]. The Cronbach’s alpha for the current sample was α = 0.87.

#### 2.3.4. Anxiety (GAD-7)

General Anxiety Disorder-7 (GAD-7) measures general anxiety [34] and is validated in German [35]. The questionnaire consists of seven four-point Likert-scaled items. Each item ranges from 0 (not at all) to 3 (nearly every day). Thus, sum scores between 0 and 21 are possible. Cut-offs of ≥11 in 14- to 17-year-olds and ≥10 in 18- to 20-year-olds were used to indicate clinically relevant anxiety symptoms [35,36]. Cronbach’s alpha for the GAD-7 was α = 0.87 in the current sample.

#### 2.3.5. Insomnia (ISI)

The Insomnia Severity Index (ISI) asks about sleep-related problems. The questionnaire is validated in German [37] and consists of seven five-point Likert-scaled items from 0 (e.g., no problem) to 4 (e.g., very severe problem). Participants can reach a maximum score of 28. Insomnia was considered if the ISI scores were ≥15 [38]. Based on the sample data, Cronbach’s alpha was α = 0.83 for the ISI.

#### 2.3.6. Perceived Stress (PSS-10)

To assess the perceived stress level of the students, the Perceived Stress Scale (PSS-10) was used. The PSS-10 contains ten items on a five-point Likert scale. The items range from 0 (never) to 4 (very often; maximum score 40). Higher perceived stress is indicated by higher sum scores. The German version is validated for adolescents above the age of 14 [39]. High levels of stress were defined by PSS-10 scores ≥27 [39,40]. Cronbach’s alpha was α = 0.86 in the studied sample.

#### 2.3.7. Eating Disorders (SCOFF)

The SCOFF screening questionnaire was used to detect eating disorders and consists of five questions addressing the core features of bulimia nervosa and anorexia nervosa. The questions can be answered with “no” (0) or “yes” (1) and disordered eating is defined by a total score ≥2 [41]. The questionnaire is validated in German [42]. Cronbach’s alpha for the current sample was α = 0.56.

#### 2.3.8. Alcohol Abuse (CAGE)

The validated screening tool CAGE [43] assesses problematic alcohol use. Its four questions target signs of alcoholism and can be answered with yes/no. Alcohol abuse was defined as a total score ≥2 [44], indicating that the participant answered two or more questions with “yes”. Based on the students’ data on hand, Cronbach’s alpha was α = 0.55 for the CAGE.

#### 2.3.9. Sociodemographic Characteristics

Participants were asked to state their age (in years). They also had to indicate their gender by clicking “female”, “male”, or “diverse”. Diverse indicated that those students did not perceive their gender to fit the binary gender norms. Furthermore, the students provided information about the federal state they live in (Burgenland, Lower Austria, Vienna, Carinthia, Styria, Upper Austria, Salzburg, Tyrol, and Vorarlberg) as well as about the type of school they attend (no school, middle school, polytechnical school, vocational school, school for intermediate vocational education, college for higher vocational education, academic secondary school, or other).

### 2.4. Data Analysis

#### 2.4.1. Qualitative Data Analysis

As a first step, responses to the open-ended question about the type of support school students wish for to improve their psychological well-being were analyzed using qualitative content analysis. A total of 245 students answered this question about the type of support they wanted with varying levels of detail. While some students only used single keywords to express their support wish, others addressed it in complete sentences.

One coder (M.W.) read through all answers to familiarize herself with the material and to get a feeling of the different types of responses. Afterward, an initial category system was created, which included main categories and thematically related subcategories. Responses were initially but not yet finally coded using the software ATLAS.ti version 22.2.5 [45]. Statements could be assigned to several categories at the same time. After the first coding phase, the category system was revised to make the categories more sharply distinguishable from each other and checking for potential allocation errors. While the initial process of coding was performed by one researcher, discussion and review of code assignments were conducted jointly with a second researcher (A.J.). Subsequently, a third researcher (T.G.), who had not been involved in the data analysis until then, coded a subset of about one-quarter (*n* = 60) of the responses. He used the category system and the definitions of the categories which were already in place. The Percentage Agreement (PA), functioning as a measure for the inter-coder agreement, was PA = 0.92, which indicated that the coders’ allocations were matching in 92% of the cases. Inconsistencies in the coding were dealt with by adding new categories to the system or clarifying coding rules.

#### 2.4.2. Quantitative and Mixed Methods Analysis

Chi-squared tests (χ^2^) or Fisher Exact Tests were used to examine:differences in the amount of clinically relevant mental health problems (depression, anxiety, stress, insomnia, eating problems, alcohol abuse) between female, male, and diverse participants, as well as between the age groups of 14 to 17 and 18 to 20 years.differences between the students with a wish for support and students without a wish for support in gender, age group, and clinically relevant mental health problems (depression, anxiety, stress, insomnia, eating problems, alcohol abuse).differences between the students with a wish for professional help and students without a wish for professional help in gender, age group, and clinically relevant mental health problems (depression, anxiety, stress, insomnia, eating problems, alcohol abuse).differences between the students with a wish to talk to someone and students without a wish to talk to someone in gender, age group, and clinically relevant mental health problems (depression, anxiety, stress, insomnia, eating problems, alcohol abuse).

Analysis (1.) represents an extended examination (high stress, eating disorders, alcohol abuse) of gender differences (male, female, and diverse) in students’ mental health to Kaltschik et al.’s [22] matched pairs analysis between girls and boys regarding depressive symptoms, anxiety symptoms, sleep problems, and perceived moderate stress. Only two of the categories that emerged from the qualitative content analysis—“professional help” for analysis (3.) and “someone to talk to” for analysis (4.)—qualified for running statistical tests since their frequencies exceeded the threshold of *n* > 30 (“professional help”, *n* = 124; “someone to talk to”, *n* = 64). The significance level for all tests was set at 5%. All tests were performed two-tailed.

## 3. Results

### 3.1. Sample

From a total of 932 participants clicking the survey link, 44 were excluded due to missing informed consent, 221 due to missing data, 21 due to being under the age of 14 or over the age of 20 years, 7 due to not attending school, and 23 since they failed the attention check question. The final sample consisted of 616 adolescents and young adults, whose ages ranged from 14 to 20 years (M = 16.63, SD = 1.49). With regards to gender, 77.4% (*n* = 477) identified as female, 19.8% (*n* = 122) as male, and 2.8% (*n* = 17) as diverse.

### 3.2. Qualitative Results

At the end of the online questionnaire, participants were asked if they would like to receive support to improve their psychological well-being. A total of 46.6% (*n* = 287) of all 616 respondents declared a need to receive support, 42 of whom did not elaborate on what type of support. A total of 245 participants answered the open-ended question in more detail. Qualitative data analysis resulted in 12 categories, which are depicted in Figure 1.

The most frequently reported category “professional help” was mentioned by 124 (20.1%) of all 616 participants and 43% of those who requested support. This category referred to psychological, psychotherapeutic, and psychiatric support, as well as to medication. In this category, we subsumed the students’ expressed desire to consult or talk to an expert, statements about hospitals or other facilities and institutions they could get in touch with as well as statements about receiving treatment or therapy. Respondent 63′s statement is an example of how students expressed their wish for professional help: *“I would like to go to a psychologist and talk everything out of my head. I want to talk about everything that went wrong in my life. I want professional help and not just the “help” of friends. I feel empty and break down almost every night and cry for half an hour. Only two people know about it and only one is in my close environment. I just want to get help, but I am afraid to talk to my mother because the last time she said to wait a little longer”*.

Participants mentioned that access to professional help should be easier, non-committal, of lower threshold, and more convenient. Waiting times should be shorter and services should be more affordable, free of charge, or paid for by health insurance. Furthermore, respondents expressed a desire for school psychologists whom they could consult more frequently or who provided appointments during breaks or class-free periods. Respondent 269, for example, expressed: *“It would be very helpful if there was a psychologist or a social worker in my school to just talk to someone where you can be sure that they will listen to you”*. Students also suggested an annual mandatory consultation: *“Just as it is obligatory to see a school doctor every year, every student should see a school psychologist at least once a year. The offer is unfortunately limited at many schools and many would not go on their own accord”* explained Respondent 144. Moreover, some participants expressed a wish for facilitated communication between their school and professional help provider.

The second most appeared category “someone to talk to” (10.4% of all participants, *n* = 64) referred to outside third parties without special expertise, people who were important to the respondents, like friends or family, as well as teachers or other types of people whom they could contact in school or in general. Often, they also explicitly wished for a person they did not know, who did not fall into their circle of family or friends (e.g., Respondent 142: *“Another person besides family and friends who listens to me talk.”*). More explicitly they expressed a desire for someone who is understanding, has a neutral perspective, whom they can trust, who does not judge them, and whom they can talk to about sensitive topics. By having such a person in their lives, they hoped to feel less pressured, receive help relaxing, reach their goals, find solutions, receive advice, or simply share their thoughts (e.g., Respondent 390: *“Someone I can confide in, so I don’t have to carry it around alone, release pressure, talk about the past”*). Moreover, students expressed the desire to obtain motivation from talking to someone or that they simply wished to know that there was another person whom they can talk to if needed. Respondent 665 furthermore stated that it would help if parents would not be informed about them talking to someone.

Furthermore, the qualitative content analysis revealed that some students desired “counseling” (2.9%, *n* = 18). This category subsumes wishes for receiving tips or advice. Participants expressed their wish for a low-threshold demand-oriented service. Furthermore, it was mentioned that a framework should be provided to reflect on one’s life situations and that understanding and support on the part of the counselor were desired. The areas in which counseling was desired varied greatly. Students hoped that their fears about the future would be alleviated, that they would be supported in making decisions about their school careers, and that counseling would generally bring clarity into their lives. Moreover, stress-relief, relief of negative thoughts, and support during difficult situations were mentioned as reasons. There was also a desire for counseling to address mental health, whether through education about symptoms indicating poor mental health, advice about mental illness, or general tips on how to feel better.

The category “information” (2.8%, *n* = 17) includes statements about getting educated, for example, about mental health, stress, and a healthy diet in form of workshops, courses, or information sheets. Respondent 314, for example, mentioned “*Perhaps information sheets, with information on how to get some distraction from these stressful school days*”. Another student (Respondent 771) asked for *“Better mental health education in school (including concerns about eating behaviors)”*.

Some participants (2.6%, *n* = 16) wished for “help with eating disorder or weight”. In this category, wishes for help with an eating disorder or abnormal eating behavior, as well as body dysmorphia, were subsumed. Some students named the desire to eat less, have an exercise partner, or someone who created a diet plan for them, so they would stop gaining weight. However, the category also referred to the wish for education on healthy eating behavior and body positivity. Participants expressed that they wanted to “*learn more about healthy nutrition. What diet suits me and my body*” (Respondent 278) or “*overcome my eating disorder, no matter what*” (Respondent 184). Some students asked for professional help in dealing with their eating disorders. Such cases were coded as both “professional help” and “help with eating disorder or weight”.

“Less pressure to perform and/or demand at school” (2.6%, *n* = 16) also emerged as an important support desire for some students. This category referred to statements that the school system and the requirements in school were stressful and students wished for less demand at school. They expressed their desire for having more free time and time for recreational activities, fewer exams, a different grading system, and a decreased school-related workload. Furthermore, criticism was voiced about the school system and the curriculum being outdated and that these should be adjusted to the current situation, which can, for example, be gathered from Respondent 10′s statement: *“The curriculum needs to be adapted to today’s generation—it is partly very outdated. The Corona measures have influenced our lives and many things are no longer the way they used to be. Young people must be supported because the only thing we are currently experiencing is psychological damage”*.

A few students (2.4%, *n* = 15) voiced that they would like to receive “support from an educational institution”. This wish for support from an educational institution contained statements about wishes for support from the school in general or their teachers more specifically. Respondents wanted teachers and other school representatives to take the mental health of their students more seriously, for example, by employing more school psychologists and educating teachers on mental health issues (e.g., Respondent 544: *“Competent school psychologists and teachers who care about mental health”*). Teachers should also be able to take actions in calls for help and receive the necessary training. Furthermore, students wished for more guidance teachers and on a regular and voluntarily basis offers to talk to health experts at school. In addition, there was a desire to be better informed about mental health and to destigmatize mental illness, especially in school settings. Besides, wishes were expressed for possibilities to relax in school and preventive measures against bullying. As with all categories, the possibility of double coding also exists within this category. For example, mentions of school psychologists were coded as both “support from an educational institution” and “professional help”.

“Family and/or friend support” (2.3%, *n* = 14) was also a support wish that some school students mentioned. Thereby they expressed their wish for family members and/or friends to support them in different aspects of their lives. For example, they had a desire to talk to them about their problems or concerns and explicitly wished for them to listen. Some students expressed the wish not to be excluded (e.g., Respondent 901 *“Friends who listen more carefully and who integrate rather than exclude me”*).

Furthermore, the category “consideration and/or understanding of others” (2.1%, *n* = 13) emerged from the content analysis. Statements in this category referred to the students’ wishes for more consideration and being treated with understanding and empathy by the people in their lives (e.g., parents, partners, teachers, peers). They wished for openness, acceptance, and not being judged both in the school context and in everyday life. They, for example, expressed *“More empathy on the part of the schools”* (Respondent 77), *“More understanding for “bad” days and a lack of drive”* (Respondent 405), or *“More consideration from teachers”* (Respondent 6).

“Destigmatization” also was a topic of importance for some participants (1.8%, *n* = 11). Statements that were subsumed under this category referred to the destigmatization of mental illnesses. They asked for mental disorders and help-seeking to be normalized in society, to no longer be taboo topics, and that people accept these to be nothing to be ashamed of. In general, they mentioned that people should be able to talk more openly about mental illnesses and receiving professional help as well as going to therapy or receiving similar support should not harm one’s career.

Some participants (1.8%, *n* = 11) did not elaborate on the type of support they wished for, but rather on the areas in which they desired help. These statements were categorized under “other support” and appeared in different areas like private problems, financial difficulties, trouble sleeping, schoolwork, or motivation for studying.

Two students (0.3%) indicated that they wished for support, but either did not know what kind of support would help them or were unable to articulate their support wish and therefore said “I don’t know”. Another 42 participants indicated that they wanted to receive support but did not answer the open-ended question about the type of support they wished for.

### 3.3. Quantitative and Mixed Methods Results

#### 3.3.1. Mental Health Indicators

Students who identified as female or diverse were more likely to show clinically relevant symptoms in five out of six mental health indicators (depression, anxiety, insomnia, stress, eating disorder) than those who identified as male. The calculated Chi^2^ tests, depicted in Table 1, showed that these five measures resulted in significant gender differences. Whereupon, clinically relevant alcohol abuse, which was represented more in male students, did not result in a significant gender difference. Bonferroni-corrected post hoc pairwise comparisons revealed that significant gender differences arose between male and female students in clinically relevant depression, anxiety, insomnia, eating disorders, and high stress. Furthermore, significant differences were found between male and diverse students in regard to clinically relevant depression, anxiety, and high stress. The post hoc tests revealed no significant differences in the mental health screenings between female and diverse adolescents (see Table 2).

The Chi^2^ tests also showed that students aged 14 to 17 were significantly more likely to show clinically relevant symptoms of anxiety and insomnia than students aged 18 to 20 years, while there was no significant age group-related difference in regard to depression, high stress, disordered eating, and alcohol abuse (see Table 1).

#### 3.3.2. Wish for Support

Furthermore, significant gender differences arose in students’ wish for support to improve their psychological well-being (see Table 3). Their wish for support was indicated by clicking “yes” on the wish for support question (see Section 2.3). Of the total, 69% of students who identified as diverse, 51.4% of the female students, and 28.9% of the male students expressed a wish for support by clicking “yes” on the support wish question. Bonferroni corrected post hoc pairwise comparison showed significant differences in the expression of a wish for support between male and female students (χ² = 19.409, *p* < 0.001) and between male and diverse students (χ² = 10.049, *p* = 0.002).

Furthermore, seventy percent of the 14–17-year-olds indicated a wish for support, while thirty percent of the 18–20-years-olds did so. Additionally, significant differences were found in the students’ wish for support in five out of six mental health indicators. As depicted in Table 3, significantly more clinically relevant mental health problems (except for alcohol abuse) were found in the sample with a wish for support compared to the sample without a wish for support.

#### 3.3.3. Types of Support Wishes

By adding the qualitative categories to the statistical analysis using a mixed methods approach, significant differences in age group and clinically relevant depression, anxiety, and stress were present between those with a wish for professional help and those with no such wish. Students with a wish for professional help were significantly more often of younger age (14–17 years) and significantly more often showed clinically relevant depression, anxiety, and high stress. As indicated in Table 3, no differences between those two groups were found for gender, insomnia, disordered eating, and alcohol abuse (although a tendency was observed here).

Furthermore, using a *Chi^2^* test, a significant difference in age and clinically relevant eating disorder was found between the group of students who expressed a wish for someone to talk to and those who did not. Students who wished for someone to talk to significantly more often belonged to the age group of 14–17-year-olds and significantly more often showed clinically relevant disordered eating than those with no such wish. As can be seen in Table 3, no significant differences between those two groups of students were found for gender and the remaining five mental health indicators.

## 4. Discussion

The present study aimed to better understand the wishes for support of Austrian school students in the light of their mental health, gender, and age. Worse mental health, specifically depression, anxiety, insomnia, eating disorders, and high stress, was significantly more prominent in adolescents with a desire for support. Of those with a wish for support, 43.2% (*n* = 124; 20.1% of the total sample), expressed their desire for professional help, which is similar to the professional support wish of school students in 2021 in Austria (47%) [46]. Clinically relevant symptoms of depression, anxiety, and high stress were significantly more prevalent among adolescents with a desire for professional help. In addition, we found out that 22.3% of those with a support wish (*n* = 64; 10.4% of the total sample) had a wish for someone to talk to. Disordered eating was significantly more prevalent in students who wished to talk to someone than those with no such wish.

This great desire for psychological and social support among students is not surprising considering the current mental health state of children and adolescents and the psychosocial care situation in Austria. Even though for years attempts have been made to expand psychosocial services in Austria [47], deficits in this care system are still apparent [48]. In 2021, only 181 school psychologists were available to cover the needs of around 1.1 million students, and in addition to counseling students, they were also responsible for teachers, parents, and supervisors [49,50]. Even before and during the early phase of the pandemic, a shortage of child and adolescent psychiatrists was already evident [51,52,53,54]. In 2010, for Austria, there was a 50% deficit in the coverage of child and adolescent psychiatry, based on the recommended number of beds set by the Federal Ministry of Social Affairs, Health, Care and Consumer Protection in the Austrian Care Health Structure Plan (Österreichischer Strukturplan Gesundheit; ÖSG) [55]. In 2022, almost three years into the pandemic and with the awareness of severely deteriorated mental health conditions, the care landscape for mentally ill children and adolescents is still inadequately equipped. There is a widespread lack of health insurance-funded psychotherapy for children and adolescents and the expansion of low-threshold services in schools is lagging [56]. The number of child and adolescent psychiatric outpatient clinics is estimated to still have a deficit of over 50% in 2022 (target number: 36, actual number: 13) [54]. In addition to the already inadequate supply situation, the increase in the prevalence of mental illnesses and the increased uptake of treatment places during the pandemic must also be taken into account, which puts further pressure on the supply situation [57].

Psychotherapeutic treatment can only be provided to 0.8% of the Austrian population [58]. There are month-long waiting times for therapy places financed by health insurance funds. In general, waiting times vary between one to six months, depending on where people live [59]. Children and adolescents must wait around 4 months for a psychotherapeutic treatment. Furthermore, to cover the demand for psychotherapeutic treatment for children and adolescents, the annual contingent of psychotherapy sessions would require an increase of 107,000 h [60]. Although there are 1204 psychotherapists with additional qualifications in infant, child, and adolescent psychotherapy, the availability for fully-financed treatment through health insurance funds comes down to five [61]. The disparity between young people’s desire for support and the actual availability of that support has to be considered a crucial driver for their mental illness and must urgently be given greater consideration in the planning of the mental health care landscape for young people in the future.

This study also showed that the prevalence of clinically relevant symptoms in all mental health measures (depression, anxiety, insomnia, high stress, and disordered eating), except for alcohol abuse, was lower in male than in female students or students who identified as diverse. The evidence of better mental health in male students was in line with other studies on gender differences in adolescents during the COVID-19 pandemic [22,23,62,63,64,65,66,67,68] reporting more symptoms of depression [62,64,65,67,68], anxiety [62,65,68], and sleeping problems in girls [66] as well as a more pronounced decline (relative to before the pandemic) in girls’ general mental well-being than in boys’ [63]. Psychosocial support directed at girls and gender-diverse individuals should therefore be given additional focus.

Compared to female and diverse students, male students were less likely to express a wish for support. On the one hand, one could argue that this was due to their better mental health. On the other hand, men’s lower rate of seeking help has been reported in numerous studies [69,70,71,72]. A systematic review of the barriers to male help-seeking behaviors, including studies from all around the world with different cultural backgrounds, pointed out that men’s reluctance to seek professional help had mainly to do with their adherence to traditional masculine norms [73]. Restricted emotional expression, the belief that men were not supposed to talk about their feelings, as well as that they were expected to cope with their problems on their own were identified as major barriers for men to seeking help [73,74,75,76,77,78,79,80,81,82]. A further barrier to seek help was the need for control and independence [73,77,78,83,84], which men seemed to see as fundamental parts of their masculine self-concept [85]. Furthermore, men often saw symptoms as minor or insignificant [73,86,87] or had the belief of being able to control the symptoms themselves or wanted to wait for them to disappear on their own [78]. Those extracted barriers were strongly associated with the fear of losing their masculinity and the desire to sustain their traditional masculine orientation. In this context, showing emotions was still seen as weak and vulnerable by many while enduring pain was perceived as being strong and resilient [73]. This could be an explanation for why male students expressed their wish for support less frequently even though many showed clinically relevant symptoms of mental illnesses.

In the group of students with no wish for professional help and those with a wish for professional help, no significant difference between clinically relevant eating disorder was found, but the group of students without a wish for someone to talk to showed significantly lower numbers of clinically relevant eating disorder compared to those with a wish for someone to talk to. This seems to be in line with what is already known about help-seeking behaviors in individuals with eating disorders. Research shows that people with disordered eating often have problems with acknowledging their illnesses’ severity and show poor mental health literacy [88,89,90], which results in not classifying their eating behaviors and symptoms to require professional treatment [91,92,93]. Furthermore, in addition to the people with eating disorders themselves, even the general public often perceived eating disorders to be conscious lifestyle choices rather than serious mental illnesses with a need for professional help [94]. Moreover, studies pointed out a reluctance to reveal information about their eating disorders [91,95]. Ali et al. [96] revealed in their systematic review that the most prominent barriers to seeking professional help in people with eating disorders were stigma and shame, failed or denied perception of the severity of their illnesses, and the fear of losing control over their eating disorder as well as being afraid of change. Especially in individuals with eating disorders, the motivation to change was found to be low [97] and the fear of having to give up the positive aspects of the illness (e.g., control, comfort, mood regulation) rather high [89]. These aspects might explain why students with clinically relevant eating disorder symptoms indeed expressed a wish for support but leaned towards talking to non-professionals, which is in line with the findings of Tavolacci et al. [98] who found friends and family members to be the two key resources for seeking help in emotionally stressful situations for individuals with bulimic eating disorder, hyperphagic eating disorder, and restrictive eating disorder.

This study again highlights the need to improve care for adolescents with mental illness. We propose that especially in the field of education, the school as a living environment makes it possible to easily reach many students with psychosocial support. Therefore, we argue that it is important to push the implementation of multi-professional health teams in educational contexts. Starting from kindergarten until students leave the educational system, children, and adolescents should be provided with preventive measures as well as acute care services according to their individual in-the-moment needs [50]. Especially for boys and young men, who are reluctant to seek professional help, and for young people with eating disorders, targeted group-specific services are particularly important. Counseling and information can be provided in an easily accessible way in the school context. Furthermore, low-threshold access to professional psychosocial help, counseling, and information should be further facilitated. If needed, therapy should be provided regionally, in a timely manner, and free of charge for students and their families. Moreover, it is central to secure the funding for psychotherapeutic treatment from health insurance to relieve the overloaded inpatient care in the medium term [50]. Consequently, an improved psychosocial infrastructure should make it possible for children and adolescents to meet their wishes for professional help.

The students’ desire for having “someone to talk to” could be potentially met by teachers whom they could confide in. They could provide space for students to talk about their problems in an everyday school setting (e.g., brief consultations during breaks or class) [99]. Furthermore, different forms of peer support (e.g., self-help groups, one-to-one mentoring; see [100]) can not only be helpful interventions for those who find informal help sufficient and “just” want someone to talk to, but also for those students for whom the barriers to seeking professional help are too big to overcome [101,102], as turning to their peers to receive support has been shown to increase students’ mental well-being [103].

### Limitations

Some limitations of this study need to be addressed. First, the collection of the data took place within an online survey framework, which especially impacted the qualitative data because students answered the open-ended questions in varying degrees of detail. Differences in the length and precision of responses became apparent during the qualitative analysis. Second, although the qualitative analysis of the types of support wishes gave an overview of how the students wished to be supported, it could not be ruled out that some support wishes were missing or underrepresented due to potentially not being on the students’ minds while filling in the online questionnaire. Nevertheless, it can be assumed that the present overview contains the most important types of support wishes. Furthermore, we have to acknowledge that in this study female students were overrepresented (female: 77.4%, *n* = 447; male: 19.8%, *n* = 122; diverse: 2.8%, *n* = 17), which means that the types of support wishes cannot be considered gender representative. Moreover, the reasons for certain support wishes as well as why some students did not wish to be supported at all remain unclear. Qualitative interviews would be suited to further examine the reasons behind why students do or do not wish to receive support. The most serious shortcoming is the convenience sample, which not only resulted in a higher percentage of girls participating but which could also have led to a self-selection bias towards students with higher mental health problems. A representative sample would have more strengths but was not possible. This has to be considered when interpreting the results. Additionally, the low internal consistencies of the CAGE (Cronbach’s α = 0.55) and the SCOFF (Cronbach’s α = 0.56) must be addressed as limitations as these raise concerns about the measures’ reliabilities.

## 5. Conclusions

Almost half of the surveyed students explicitly expressed a wish for receiving support to improve their psychological well-being. Significantly more of those who desired support also showed poorer mental health in terms of depression, anxiety, eating disorders, insomnia, and high stress than without a support wish. Most students expressed their wish for support by asking for professional help. Considering young people’s mental health and their explicit wish for help, we propose facilitated access and financial support for psychosocial services, especially for young people, and educative measures to improve adolescents and young adults’ mental health literacy.

## Figures and Tables

**Figure 1 ijerph-20-04749-f001:**
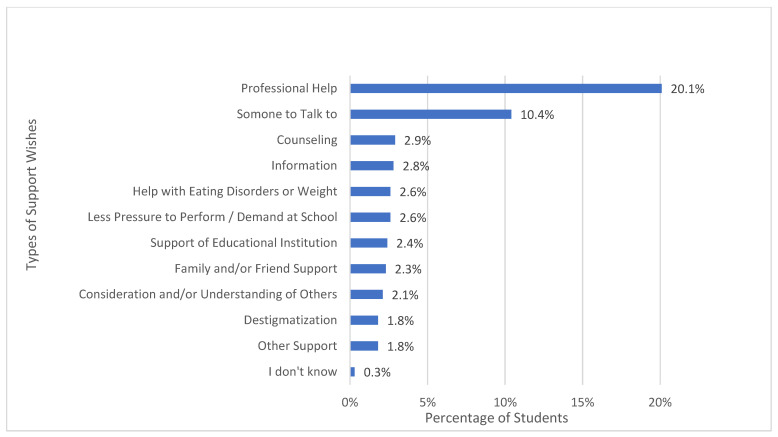
Percentages of school students who mentioned a certain category of support. Statements could be assigned to several categories at the same time.

**Table 1 ijerph-20-04749-t001:** Mental health indicators by gender and age.

	Gender				Age		
	Female	Male	Diverse	Statistics	14–17 Years	18–20 Years	Statistics
*N*	477	122	17		439	177	
Depression, %	72.7%	45.1%	94.1%	χ²(2;616) = 39.616;	66.5%	71.2%	χ²(1;616) = 1.262;
(*n*)	347	55	16	*p* < 0.001	292	126	*p* = 0.261
Anxiety, %	57.0%	35.2%	70.6%	χ²(2;616) = 20.106;	51.3%	62.7%	χ²(1;616) = 6.681;
(*n*)	279	45	12	*p* < 0.001	225	111	*p* = 0.010
Insomnia, %	33.5%	20.5%	41.2%	χ²(2;616) = 8.530;	28.7%	37.3%	χ²(1;616) = 4.335;
(*n*)	160	25	7	*p* = 0.014	126	66	*p* = 0.037
High Stress, %	43.4%	22.1%	64.7%	χ²(2;616) = 22.877;	40.3%	38.4%	χ²(1;616) = 0.190;
(*n*)	207	27	11	*p* < 0.001	177	68	*p* = 0.663
Disordered Eating, %	56.9%	32.2%	43.8%	χ²(2;608) = 23.869;	52.4%	47.5%	χ²(1;608) = 1.307;
(*n*)	268	39	7	*p* < 0.001	230	84	*p* = 0.235
Alcohol Abuse, %	16.8%	21.5%	11.8%	χ²(2;615) = 1.889;	16.2%	20.9%	χ²(1;615) = 1.918;
(*n*)	80	26	2	*p* = 0.389	71	37	*p* = 0.166

Note. Mental health indicators (depression, anxiety, insomnia, stress, disordered eating, alcohol abuse) were dichotomized (not clinically relevant; clinically relevant) according to established cut-offs. For depression (PHQ-9) a cut-off point ≥10 was used in participants aged 18 or older [31] to define clinically relevant depressive symptoms, whereas a cut-off ≥11 was used for adolescents aged between 14 and 17 [32]. For anxiety symptoms (GAD-7), cut-offs ≥11 in 14- to 17-year-old adolescents and ≥10 in 18- to 20-year-old adolescents were used [35,36]. Insomnia was considered if ISI scores were ≥15 [38]. High stress levels were defined by PSS-10 scores ≥27 [40]. Alcohol abuse (CAGE) [44] and disordered eating (SCOFF) [41] were defined as total scores ≥2.

**Table 2 ijerph-20-04749-t002:** Post hoc pairwise gender comparisons of the mental health screenings.

	Male-Divers		Female-Divers		Female-Male	
	χ²	*p*	χ²	*p*	χ²	*p*
Depression (PHQ-9)	14.358	**<0.001**	3.848	0.050	33.686	**<0.001**
Anxiety (GAD-7)	7.006	**0.008**	0.992	0.319	18.262	**<0.001**
Insomnia (ISI)	3.602	0.058	0.427	0.513	7.752	**0.005**
Eating Disorders (SCOFF)	0.841	0.359	1.089	0.297	23.466	**<0.001**
High Stress (PSS-10)	13.615	**<0.001**	3.023	0.082	18.456	**<0.001**

Note. *p*-values that remained significant after a familywise Bonferroni correction are displayed in bold.

**Table 3 ijerph-20-04749-t003:** Wish for support, professional help, and someone to talk to by gender, age group, and mental health indicators.

	Support Wish			Professional Help			Someone to Talk to		
	Yes, %	*n*	Statistics	Yes, %	*n*	Statistics	Yes, %	*n*	Statistics
Gender			χ²(2;606) = 22.479 *p* < 0.001			χ²(2;616) = 4.290 *p* = 0.117			Fisher’s Exact Test *p* = 0.115
Female	51.4%	241		21.4%	102		11.9%	57	
Male	28.9%	35		13.9%	17		5.70%	7	
Diverse	68.8%	11		29.4%	5		0.0%	0	
Age			χ²(21;606) = 0.315 *p* = 0.575			χ²(1;616) = 6.375 *p* = 0.012			χ²(1;606) = 4.650 *p* = 0.031
14–17 years	70.0%	201		62.1%	77		82.8%	53	
18–20 years	30.0%	86		37.9%	47		17.2%	11	
Depression (PHQ-9)			χ²(1;606) = 50.415 *p* < 0.001			χ²(1;616) = 14.762 *p* < 0.001			χ²(1;616) = 0.529 *p* = 0.467
Below cut-off	26.5%	52		11.1%	22		9.1%	18	
Above cut-off	57.3%	235		24.4%	102		11.0%	46	
Anxiety (GAD-7)			χ²(1;606) = 44.406 *p* < 0.001			χ²(1;606) = 32.763 *p* < 0.001			χ²(1;616) = 0.256 *p* = 0.613
Below cut-off	32.5%	89		10.0%	28		11.1%	31	
Above cut-off	59.6%	198		28.6%	96		9.8%	33	
Insomnia (ISI)			χ²(1;606) = 10.544 *p* = 0.001			χ²(1;616) = 2.543 *p* = 0.111			χ²(1;616) < 0.001 *p* = 0.988
Below cut-off	42.9%	179		18.4%	78		10.4%	44	
Above cut-off	57.1%	108		24.0%	46		10.4%	20	
Eating Disorders (SCOFF)			χ²(1;606) = 14.968 *p* < 0.001			χ²(1;608) = 0.998 *p* = 0.318			χ²(1;608) = 4.417 *p* = 0.036
Below cut-off	39.2%	115		18.7%	55		7.8%	23	
Above cut-off	55.0%	172		22.0%	69		13.1%	41	
High Stress (PSS)			χ²(1;606) = 61.971 *p* < 0.001			χ²(1;616) = 32.300 *p* < 0.001			χ²(1;616) = 0.915 *p* = 0.339
Below cut-off	34.3%	125		12.7%	47		9.4%	35	
Above cut-off	66.9%	162		31.4%	77		11.8%	29	
Alcohol Abuse (CAGE)			χ²(1;606) = 2.443 *p* = 0.118			χ²(1;615) = 3.642 *p* = 0.064			χ²(1;615) = 0.374 *p* = 0.541
Below cut-off	45.9%	229		18.7%	95		10.1%	51	
Above cut-off	54.2%	58		26.9%	29		12.0%	13	

Note. Mental health indicators (depression, anxiety, insomnia, stress, disordered eating, alcohol abuse) were dichotomized (not clinically relevant; clinically relevant) according to established cut-offs. For depression (PHQ-9) a cut-off point ≥10 was used in participants aged 18 or older [31] to define clinically relevant depressive symptoms, whereas a cut-off ≥11 was used for adolescents aged between 14 and 17 [32]. For anxiety symptoms (GAD-7), cut-offs ≥11 in 14- to 17-year-old adolescents and ≥10 in 18- to 20-year-old adolescents were used [35,36]. Insomnia was considered if ISI scores were ≥15 [38]. High stress levels were defined by PSS-10 scores ≥27 [40]. Alcohol abuse (CAGE) [44] and disordered eating (SCOFF) [41] were defined as total scores ≥2. Wish for support was indicated by clicking “yes” on the wish for support question (see Variables).

## Data Availability

The datasets generated during the current study are available from the corresponding author upon reasonable request.

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
