# Peer review of "An Assessment of Austrian School Students’ Mental Health and Their Wish for Support: A Mixed Methods Approach"

_ijerph, 2023, doi:10.3390/ijerph20064749_

Round 1
Reviewer 1 Report
This is a good piece of research. This exploratory study is well founded and follows a robust methodological path. Findings are relevant for educational and mental health policy. Some suggestions to improve this piece: Avoid mention COVID, the time of the study is after the pandemic, and we do not want cause-effect presumptions. Findings stand alone as a diagnosis of potential mental health issues regardless of COVID. Figure 1 must be arranged horizontally for clarity. Table 1 is unnecessary since differences within are consistent and they can be simply commented in a paragraph (table 2 stands because of the differences). Good Job.
Author Response
Dear reviewer,
thank you very much for reviewing our paper and providing valuable comments to improve it. In the following, we would like to briefly address your comments:
- While we agree with your point that the results are also an important finding for student mental health independent of Corona, we still believe that the Corona pandemic has had and continues to have a major impact on student mental health and their desire for support. The ongoing reference to Corona is also evident in some responses to the open-ended questions about the type of support desired (see, for example, lines 331-334). In addition, the Corona pandemic in Austria is, even to date, still ongoing (expected to end by the end of June this year), which is why we believe it is important for readers to understand the temporal context in which the data were collected and the socio-political events that impacted daily life during this time. We thus view the Corona pandemic as important contextual information and, contrary to your valued advice, have decided against omitting mentions of COVID-19.
- As recommended, we arranged Figure 1 horizontally for more clarity.
- Since we think that tables offer the advantage of being able to better compare the statistical values side by side and because Reviewer 2 has asked to calculate post hoc tests for the differences between the three sexes, in the course of which we refer to this table, we decided to keep the table in the paper.

Reviewer 2 Report
In general, the introduction was well-written.
1. Need to include the available evidence on this from other countries to establish whether the gap is common or local.
2. The last sentence in Introduction (Hence, the study explores...) need to be reworded as it is not clear. Consider breaking the sentence into several sentences.
3. Methodology - conflicting info - mixed method in title but cross sectional in methodology.
4. If mixed method need to explain which design (exploratory/explanatory/embedded?) and how does the data inform each other.
5. Variables - need to list the demographics asked
6. Sub 2.3.1 until 2.3.6 - Instead of mentioning Cronbach's alpha was..., authors need to include whether the instrument has been validated in their context and what was the Cronbach.
7. In view that the data analysis begin with qualitative item, suggest to explain wish for support first before other instruments.
8. What are other measures to increase trustworthiness in qualitative data analysis apart from inter coder agreement?
9. Tables - remove this ' Error! Reference source not found'
10.Table 1 and 2 (gender) - suggest authors to do post hoc test to see significant pair difference - https://alanarnholt.github.io/PDS-Bookdown2/post-hoc-tests-1.html
11. Table 2, Table 3 and Table 4. can delete column no, % and n as it is a dichotomous response. Can even merge into one table (portrait mode) to ease comparison
12. Excellent discussion.
Author Response
Dear reviewer,
thank you very much for taking the time and reviewing our paper. We very much appreciate your valuable feedback. Below we address your comments as implemented in the paper:
- Thank you for this important comment. We have made it clear that the gap is not an Austria-specific problem, but that there is a general research gap in the area of support wishes among adolescents.
- We have split the last sentence of the introduction into two to make it easier to understand.
- We believe that the information “cross-sectional” and “mixed-methods” does not conflict, since “cross-sectional” (as the opposite of longitudinal) describes the temporal focus of the study while "mixed-methods" refers to the combination of qualitative and quantitative data collection and analysis. Using mental health screenings and open-ended questions, we collected both quantitative and qualitative data at one time point. Therefore, this was a cross-sectional mixed-methods study design.
- We added that this was an exploratory-sequential mixed methods design in which the qualitative analysis informed the quantitative analysis by providing the support wishes categories.
- In addition to gender, we also described the sociodemographic information of age, federal state, and school type.
- We apologize for any misunderstandings. All of the standardized instruments used have been validated in previous studies, not only for the original language but also in German. The respective studies are cited in the methods section. In the text, we report the internal consistency (Cronbach’s alpha) calculated for the sample at hand for each of the scales to follow APA reporting standards for quantitative studies. We edited this section to improve comprehensibility by specifying that the Cronbach’s alphas referred to the sample at hand and by adding additional sources for the validation of the German versions of the questionnaire.
- As suggested, we reorganized the variable section and started by describing the wish for support since the qualitative analysis was first in order.
- By describing in detail how the data were analyzed (which researcher was involved at which point in the process, how the category system was created, how refinements were made in the category system, and how the codes were assigned), we aimed to meet the qualitative quality criterion of transparency. The quality criterion of intersubjectivity was considered in that the data analysis was not carried out by one researcher alone but within a research team. Intersubjectivity is achieved through consensual validation. In terms of the scope, we have described that this is a convenience sample and not a representative sample. Moreover, we refrained from generalizing the qualitative results anyway. In addition to the inter-coder agreement, we believe that we have thus presented the quality criteria of qualitative research in a sufficiently transparent way in the course of the paper.
- We think that this message ("Error! Reference source not found") is a display error in Word, since this error message does not appear in our version of the paper, and therefore we hope that this error also won’t be displayed anymore in the updated version of the paper that you are now receiving.
- Following your guidance, we performed post hoc pairwise comparisons to see where the differences between the genders lie. Most differences occurred between males and females, but some also between male and diverse students. Your recommendation has significantly increased the clarity of our results.
- Following your recommendation, we deleted the columns “no &, n” and merged the three tables into one to make the results easier to compare.
- Thank you for the appreciation of our work!

Reviewer 3 Report
Dear Authors,
Congratulations on your very good work. It addresses the extremely important topic of adolescent mental health. The methodology, which is very well developed, deserves attention. In this respect, I have nothing to add, as the study was excellently designed and thought out.
However, I would suggest adding analyses in relation to age - coping with different emotions and therefore problems during the COVID-19 pandemic and mental health status may differ between 14-15 year olds and older adolescents. It would therefore be worth considering such analyses.
Please also correct the title of Table 1(delete "Error! Reference source not found.").
An interesting addition to the introduction would have been a brief outline of the types of restrictions during the COVID-19 pandemic in Austria. To what extent were young people affected (e.g. were remote learning, bans on moving, meeting etc. introduced?).
Good luck.
Author Response
Dear reviewer,
thank you for your appreciative words about our work and your much-valued feedback.
Below we briefly discuss the points that have been implemented:
- We followed your suggestion and ran additional Chi2 tests to compare younger (14-17 years) and older (18-20 years) students and found significant differences in clinically relevant anxiety and insomnia symptoms as well differences between those with a wish of “professional help” and “someone to talk to” in regard to their age group.
- We think that this message ("Error! Reference source not found") is a display error in Word, since this error message does not appear in our version of the paper, and therefore we hope that this error also won’t be displayed anymore in the updated version of the paper that you are now receiving.
- As recommended, we gave a brief outline of the COVID-19 related measures in Austria. We described the measures in place at the time of data collection to provide a frame of reference as to what students were limited by during this period and also specifically addressed school closures in the course of the pandemic.

Round 2
Reviewer 2 Report
Thank you for addressing the suggestions. I have no further comments except for several minor suggestions as following
Typo: Omicron (Line 57), focused (Line 84)
Suggest to reword study objective - The study also examined gender and age group differences in the amount of clinically relevant mental health problems, and investigate differences in gender, age, and mental health between students with a general wish for support, a wish for professional help as well as a wish for someone to talk to.